# Addressed Fiber Bragg Structures in Load-Sensing Wheel Hub Bearings

**DOI:** 10.3390/s20216191

**Published:** 2020-10-30

**Authors:** Timur Agliullin, Robert Gubaidullin, Airat Sakhabutdinov, Oleg Morozov, Artem Kuznetsov, Valentin Ivanov

**Affiliations:** 1Department of Radiophotonics and Microwave Technologies, Kazan National Research Technical University named after A.N. Tupolev-KAI, 420111 Kazan, Russia; diablogrr@gmail.com (R.G.); kazanboy@yandex.ru (A.S.); microoil@mail.ru (O.M.); aakuznetsov@kai.ru (A.K.); 2Automotive Engineering Group, Technische Universität Ilmenau, 98693 Ilmenau, Germany; valentin.ivanov@tu-ilmenau.de

**Keywords:** microwave photonic sensor system, numerical simulation, addressed fiber Bragg structures, load-sensing bearings, vehicle dynamics control

## Abstract

The work presents an approach to instrument the load-sensing bearings for automotive applications for estimation of the loads acting on the wheels. The system comprises fiber-optic sensors based on addressed fiber Bragg structures (AFBS) with two symmetrical phase shifts. A mathematical model for load–deformation relation is presented, and the AFBS interrogation principle is described. The simulation includes (i) modeling of vehicle dynamics in a split-mu braking test, during which the longitudinal wheel loads are obtained, (ii) the subsequent estimation of bearing outer ring deformation using a beam model with simply supported boundary conditions, (iii) the conversion of strain into central frequency shift of AFBS, and (iv) modeling of the beating signal at the photodetector. The simulation results show that the estimation error of the longitudinal wheel force from the strain data acquired from a single measurement point was 5.44% with a root-mean-square error of 113.64 N. A prototype load-sensing bearing was instrumented with a single AFBS sensor and mounted in a front right wheel hub of an experimental vehicle. The experimental setup demonstrated comparable results with the simulation during the braking test. The proposed system with load-sensing bearings is aimed at estimation of the loads acting on the wheels, which serve as input parameters for active safety systems, such as automatic braking, adaptive cruise control, or fully automated driving, in order to enhance their effectiveness and the safety of the vehicle.

## 1. Introduction

Various active safety systems such as the antilock braking system (ABS) and electronic stability control (ESC) have been used to increase driving safety for several decades [1,2]. However, their efficiency depends on the correctness of tire–road traction parameters that are currently estimated by means of indirect methods [3,4] which do not always provide sufficient accuracy, therefore limiting the capabilities of such systems for vehicle dynamics control.

In order to mitigate this issue, a significant research effort is aimed at the development of the corresponding direct methods, in particular, based on the real-time measurement of loads acting on the wheels, using a variety of sensor types. Theoretically, every component of the vehicle transferring the loads from the tire contact patch to the vehicle body can be used to estimate the wheel loads. Thus, four main categories of wheel force determination methods can be distinguished in accordance with the vehicle components they are based on: tire-based methods (including the ones that use piezoelectric [5,6,7], fiber-optic [8,9,10], or optical [11,12,13] sensors; electrical resistance strain gauges [14,15]; or accelerometers [16]); wheel disc-based methods (primarily used in testing equipment [7]); suspension-based methods (including strain measurement of suspension bushings [17], displacement measurement of knuckle [18]), and wheel hub bearing based methods (load-sensing bearings) [19,20,21,22,23,24]. Wheel force measurements based on load-sensing hub bearings have a number of advantages in comparison with the other approaches. Firstly, the sensors are located on a nonrotating ring of the bearing, therefore significantly simplifying data acquisition. Secondly, the hub bearings generally have a much longer lifespan than tires. Thirdly, the methods based on load-sensing bearings are robust in case of tire or wheel change. Finally, load-sensing hub bearings are located closer to the contact patch than other suspension components, therefore providing a more precise load estimation.

The current work aims to present a novel approach to instrument the load-sensing hub bearings with fiber-optic sensors suitable for automotive applications. The paper introduces a comprehensive model of the load measurement process and an experimental demonstration of the system operation.

## 2. Load-Sensing Bearings in Automotive Applications

Figure 1 illustrates the dependence of wheel loads on road conditions by presenting examples of the longitudinal wheel force as the function of the wheel slip ratio *k* for various conditions of the road surface [25]. As can be seen from the figure, the magnitude of the wheel force varies for different road conditions in the wide range.

The wheel slip ratio *k* is determined using the Expressions (1) and (2) in case of braking and acceleration, respectively:(1)$$k=\frac{v_{x}-\omega_{w}r_{w}}{v_{x}},$$
(2)$$k=\frac{\omega_{w}r_{w}-v_{x}}{\omega_{w}r_{w}},$$where *v_x_* is the longitudinal velocity of the vehicle, ω_w_ is the angular velocity of the wheel, and *r*_w_ is the wheel dynamic radius.

As it is demonstrated in Figure 1, the peak value of the longitudinal wheel force is reached at a certain wheel slip ratio, which corresponds to the most efficient braking or acceleration [26]. It must be noted that the wheel slip coinciding with the maximum wheel force is different for various road conditions, which further complicates its indirect estimation. When the vehicle brakes or accelerates, the load-sensing wheel hub bearings identify the maximum wheel force, and the slip ratio is corrected in such a way as to maintain the peak value of the wheel force, therefore enhancing the effectiveness of the wheel–road interaction [24]. It should be noted that some low-friction roads as well as deformable surfaces have no clearly pronounced extremum of the longitudinal force–slip-curve. In this case, the analysis of the curve slope should be considered by seeking the peak value. However, the presented paper does not include case studies with such surfaces.

In this work, we consider wheel hub bearings with nonrotating outer rings. The load caused by the wheel–road interaction is applied to the inner ring of the bearing, and then it is translated to the outer ring by means of the rolling elements (balls) and induces strain of the outer ring. This tangential deformation detected by a single sensor is periodic and sinusoidal even at constant load due to the passage of the rolling elements during bearing rotation [27]. Therefore, the sensor detects the maximum strain when its location coincides with the ball position, and the minimum strain is detected when the two adjacent rolling elements are equidistant from the sensor [22] (see Figure 2, where *P*_r_ is the radial load applied to the inner ring of the bearing).

The current work proposes the usage of fiber-optic sensors for strain measurement of the bearing outer ring. With all the intrinsic advantages of fiber-optic sensors based on fiber Bragg gratings (FBGs), such as low weight, small footprint, immunity to electromagnetic interference, and absence of electric power supply, a significant disadvantage still persists that is related to the usage of electro-optic interrogators. The complexity and high cost of such devices, which are designed to define the FBG central frequency by scanning its spectral response, constrain the areas of application of FBG-based sensor systems [28,29].

In order to solve the abovementioned issue, the addressed fiber Bragg structures (AFBSs) have been developed [24,27,30,31,32,33,34,35] and further expanded into multi-addressed fiber Bragg structures [29]. An AFBS is a fiber Bragg grating that has the optical frequency response with two narrowband components. The frequency spacing between the components is called the address frequency and lies in the radio frequency range. The system can include several AFBSs with the same Bragg frequency, and the sensors are distinguished using their address frequencies, which are unique for each sensor in the system and do not change when the sensors are subjected to strain or temperature fields. Two approaches for AFBS design have been proposed to date: 2λ-FBG and 2π-FBG. The 2λ-FBG is based on two sequentially formed ultra-narrowband FBGs with different Bragg frequencies [30,31]. An addressed structure with two symmetrical π-phase shifts (2π-FBG) consists of three FBGs divided by the discrete symmetrical phase shifts [24,27,32,33,34]. The 2π-FBGs are generally shorter in comparison with 2λ-FBGs, which makes them more suitable for applications where minimal sensor length is required, for instance, in load-sensing bearings, due to their relatively small diameter [30].

Thus, the application of 2π-FBGs as sensing elements of load-sensing hub bearings allows mitigating the disadvantages of the traditional FBGs, such as the high cost of interrogation devices, while maintaining the benefits of fiber-optic sensor technology and making the system suitable for automotive applications.

## 3. AFBS Interrogation Principle

Figure 3 presents the optic-electrical scheme of the interrogation system for two AFBSs of 2π-FBG type. A wideband light source (1) generates light radiation (diagram a) corresponding to the range of the frequency shift of the sensors. The light passes through a fiber-optic splitter (9) and enters the 2π-FBG-sensors (2.1 and 2.2), both of which form two-frequency radiations that are merged into a four-frequency optical response (diagrams b, c) by a fiber-optic coupler (10). After that, the optical signal is divided into two channels (the measuring one and the reference one) via a fiber-optic splitter (6). The measuring channel is equipped with an optical filter (3) with a predefined linear inclined frequency response, which converts the two-frequency radiation into the asymmetrical one (diagram d). After that, the signal is guided to the photodetector (4) and is processed by the measuring analog-to-digital converter (ADC) (5). The output signal of the ADC (5) is used to determine the central frequencies of the AFBS sensors. In the reference channel, the unmodified signal (diagram e) is transmitted directly to the reference photodetector (7) and then undergoes processing by the reference ADC (8). All the subsequent calculations for AFBS central frequency determination are carried out using the relations of the signal amplitudes from the outputs of the measuring and reference ADCs [27]. Therefore, the intensity of the output signal is normalized, eliminating the influence of the light source power fluctuations on the amplitudes of the AFBSs’ optical spectrum response.

The optical response from the *i*-th AFBS is represented as follows:(3)$$E_{i}(t)=A_{i}e^{j\omega_{i}t+\varphi_{Ai}}+B_{i}e^{j(\omega_{i}+\Omega_{i})t+\varphi_{Bi}},$$where *A_i_* and *B_i_* are the amplitudes of the AFBS spectral components, ω*_i_* is the frequency of the left spectral component of the *i*-th AFBS, Ω*_i_* is the address frequency, and φ*_Ai_* and φ*_Bi_* are the phases. It must be noted that the proposed mathematical representation of the AFBS spectrum does not take into account the spectral shape of the transparency windows, which can be described using the Gaussian (in case of 2λ-FBG [31]) or Lorentz (in case of 2π-FBG [27]) functions.

The luminous power received by the photodetector from *N* addressed structures can be described using Expression (4) by multiplying Expression (3) with its complex conjugate:(4)$$\begin{array}{ll} P(t)= & \left(\sum_{i=1}^{N}E_{i}(t)\right)\left(\bar{\sum_{i=1}^{N}E_{i}(t)}\right)=\left(\sum_{i=1}^{N}\left(A_{i}e^{j\omega_{i}t+\varphi_{Ai}}+B_{i}e^{j(\omega_{i}+\Omega_{i})t+\varphi_{Bi}}\right)\right)\left(\sum_{k=1}^{N}\left(A_{k}e^{-(j\omega_{k}t+\varphi_{Ak})}+B_{k}e^{-(j(\omega_{k}+\Omega_{k})t+\varphi_{Bk})}\right)\right)=\\ & =\sum_{i=1}^{N}\left(A_{i}{}^{2}+B_{i}{}^{2}\right)+2\sum_{i=1}^{N}A_{i}B_{i}\cos(\Omega_{i}t+\varphi_{Ai}-\varphi_{Bi})+\\ & +2\sum_{i=1}^{N}\sum_{k=i+1}^{N}\left(\begin{array}{l} A_{i}A_{k}\cos((\omega_{i}-\omega_{k})t+\varphi_{Ai}-\varphi_{Ak})+\\ A_{i}B_{k}\cos((\omega_{i}-\omega_{k}-\Omega_{k})t+\varphi_{Ai}-\varphi_{Bk})+\\ B_{i}A_{k}\cos((\omega_{i}-\omega_{k}+\Omega_{i})t+\varphi_{Bi}-\varphi_{Ak})+\\ B_{i}B_{k}\cos((\omega_{i}-\omega_{k}+\Omega_{i}-\Omega_{k})t+\varphi_{Bi}-\varphi_{Bk}) \end{array}\right). \end{array}$$

As can be seen, the oscillation of the output electrical signal of the photodetector at the address frequency Ω*_i_* is proportional to the amplitudes of the AFBS optical spectral components *A_i_* and *B_i_*. The amplitudes *A_i_* and *B_i_* are defined by the parameters *u* and *v* of the linear function describing the frequency response of the optical filter with inclined frequency response ((3) in Figure 3):(5)$$A_{i}=L_{0}(u\omega_{i}+v),B_{i}=L_{0}(u(\omega_{i}+\Omega_{i})+v),$$where *u* is the slope ratio, *v* is the free term of the equation describing the frequency response of the optical filter, and *L*_0_ is the amplitude of the optical spectral component of the AFBS prior to entering the filter with inclined linear frequency response. By measuring the amplitude of the photodetector output signal at the address frequency Ω*_i_*, it is possible to define the central frequency shift (or the frequency of the left spectral component ω*_i_*) of the AFBS. However, due to the appearance of the additional frequency components in the last sum of Expression (4), the filtering of the signal at the address frequencies is required.

## 4. Modeling of Vehicle Dynamics

As mentioned before, load-sensing bearings are used to assess the tire–road friction characteristics based on the wheel forces. In order to define the forces acting on the wheels in various conditions, consider a split-mu braking test, which was simulated using the CarSim software. A split-mu braking test is a common vehicle testing procedure that presupposes straight braking in a lane with significantly different frictional coefficients for the left and the right wheel paths. The visualization of the maneuver is shown in Figure 4a.

The vehicle under test was a generic C-class hatchback with the following main parameters: a sprung mass of 1270 kg, a tire size of 215/55 R17, and a generic braking system with ABS. The friction of the left side of the road was μ_L_ = 0.2, and the friction of the right side was μ_R_ = 0.5, which imitates the possible road conditions at low temperatures. The maneuver included braking from the initial velocity of 65 km/h to a standstill with the constant pressure of 15 MPa applied to the brake master cylinder. The ABS was activated during the test in order to eliminate skidding. The profiles of the vehicle longitudinal velocity and steering wheel angle are presented in Figure 4b. As can be seen, the virtual driver applied a certain steering input in order to compensate for the yaw moment generated due to the inequality of left and right wheel forces.

The longitudinal wheel forces obtained from the simulation during the split-mu braking test are presented in Figure 5. Using these values, the corresponding deformation of the hub bearing outer ring can be calculated, which is described in the next section.

## 5. Modeling of Bearing Outer Ring Deformation

In order to estimate the tangential deformation of the outer ring of the load-sensing bearing caused by the loads acting on the wheels during the maneuver, consider a beam model with simply supported boundary conditions [23]. The model provides sufficient accuracy for preliminary estimations while maintaining simplicity of calculations [23]. A schematic representation of the beam model is shown in Figure 6, where the following denotations are used: *P* is the load transmitted to the outer ring by a ball, *P*_r_ is the radial load applied to the inner ring of the bearing, *x* is the position of the sensor (i.e., the point at which the strain is estimated), and *a* and *b* are the load positions relative to the left end and the right end of the beam, respectively.

As mentioned in Section 2, the strain of the outer ring measured by a single sensor is periodic due to the passage of rolling elements. The maximum values of strain are achieved when the position of the ball coincides with the position of the sensor, and the sensor is located in the middle of a beam, i.e., *a* = *b* = *x*, while the minimum strain is detected when the sensor is positioned in the middle between two adjacent balls, i.e., *x* − *a* = *β*/2, where *β* is the arc length between the positions of two subsequent balls. In order to define the load applied by the rolling element to the beam, we use a generally accepted relation found by Stribeck in 1900, according to which the load on the most-loaded element is 4.37 times higher than the average load distributed on the bearing balls [36]:(6)$$P=4.37\frac{P_{r}}{n},$$where *n* is the number of balls in a single row of a bearing.

A single load acting on the bearing ball in a beam model with simply supported boundary conditions is expressed as follows [23]:(7)$$P=-\frac{L}{bx}\frac{S_{x}EI}{y},$$where *S_x_* is the tangential strain at ‘*x*’ distance from the left end of a beam; *E* is Young’s modulus; *I* is the area moment of inertia of a cross-section of the beam; *y* is distance from the neutral axis, where the strain is calculated (i.e., half of the outer ring thickness); and *L* = *a* + *b* is the beam length (i.e., half of the perimeter of the outer ring circumference).

Considering *a* = *b* = *x* = *L*/2 for the maximum case and *x* = *L*/2, *b* = *L*/2 − *L*/*n* for the minimum case, the corresponding values of strain can be estimated based on Expression (7):(8)$$S_{x,\max}=-\frac{{\left(L/2\right)}^{2}}{L}\frac{yP}{EI},$$
(9)$$S_{x,\min}=-\frac{(n-2)L}{4n}\frac{yP}{EI}.$$

Using Expressions (8) and (9), the tangential deformation of the bearing outer ring was calculated for the whole duration of the maneuver. The results are presented in Figure 7a for the whole duration of the simulated testing procedure and in Figure 7b for a shorter time interval. As can be seen from the figure, the sensor on the front right hub bearing is subjected to significantly higher strain in comparison with the left one due to higher loads acting on the right wheel caused by higher road traction.

A system based on load-sensing bearings is aimed at solving the inverse problem, i.e., the calculation of force using the measured values of strain. In order to assess the estimation accuracy for a system with a single strain-sensor, an upper envelope of the simulated strain was obtained for the front right wheel hub of the simulated vehicle, which is shown in Figure 8a. After that, the longitudinal wheel force was estimated using Expressions (6) and (7) considering only the maximum strain case. The results are illustrated in Figure 8b, and the estimation error found as the difference between simulated and estimated forces is presented in Figure 8c. The high overshoot of error at the beginning of simulation results from linear interpolation of strain due to the relatively long time interval as well as rapid strain raise between the subsequent strain peaks. The root-mean-square error for the whole duration of the simulation was 113.64 N, which indicates a deviation of 5.44% from the average force of 2087.15 N.

## 6. Modeling of AFBS Interrogation

Consider a 2π-FBG structure with the address frequency Ω = 3.75 GHz as the sensing element for strain measurement of the bearing outer ring. The simulations presented in this section were performed using the OptiSystem 7.0 software. The optoelectronic interrogation scheme implemented for the simulation is shown in Figure 9. The model does not include the reference channel shown in Figure 3, since the optical source is ideal and does not induce fluctuations of signal intensity. The AFBS spectrum was obtained using the OptiGrating software, and the left slope of the frequency response of the trapezoidal optical filter with the central frequency of 193.215 THz and bandwidth of 136.979 GHz was utilized as a filter with inclined frequency response. The abovementioned filter parameters provide a linear frequency response of the filter in the whole range of AFBS sensor frequencies.

It is known that the central frequency shift of AFBS, similarly to conventional FBG, depends both on the applied strain and temperature. According to [37], the strain measurement can be represented as a function of the central frequency shift for the strain sensor and the temperature sensor:(10)$$\begin{array}{c} g(\Delta f_{T},\Delta f_{P})=c_{2,3}\cdot \Delta f_{T}^{2}\cdot \Delta f_{P}^{3}+c_{2,2}\cdot \Delta f_{T}^{2}\cdot \Delta f_{P}^{2}+c_{2,1}\cdot \Delta f_{T}^{2}\cdot \Delta f_{P}+\\ +c_{2,0}\cdot \Delta f_{T}+c_{1,2}\cdot \Delta f_{T}\cdot \Delta f_{P}^{2}+c_{1,1}\cdot \Delta f_{T}\cdot \Delta f_{P}^{2}+c_{1,1}\cdot \Delta f_{T}\cdot \Delta f_{P}+\\ +c_{1,0}\cdot \Delta f_{T}+c_{3,0}\cdot \Delta f_{P}^{3}+c_{2,0}\cdot \Delta f^{2}+c_{0,1}\cdot \Delta f_{P}+c_{0,0}, \end{array}$$where Δ*f*_P_ is the shift of the central frequency due to the deformation, Δ*f*_T_ is the shift of the central frequency due to temperature exposure, and *c_i,j_* are calibration coefficients.

In the current simulation, the temperature of the sensor is assumed to be constant and its influence on the central frequency shift is neglected. The software used for modeling the vehicle dynamics is unable to estimate the temperature of bearings; therefore, further studies are required to assess their temperature. Several papers have reported thermal modeling of bearings [38,39]. Nevertheless, due to the relatively short duration of the testing procedure, the temperature variation of the bearing is not expected to exceed several degrees Celsius, which results in the central frequency shift that is at least by one order of magnitude smaller than the one caused by the bearing strain (considering typical FBG temperature sensitivity of approximately 10 pm/°C in terms of wavelength [40]). If the temperature influence is excluded from Equation (10), then the function relating the shift of the central frequency and deformation can be represented as follows:(11)$$g(\Delta f_{P})=c_{3}\cdot \Delta f_{P}^{3}+c_{2}\cdot \Delta f_{P}^{2}+c_{1}\cdot \Delta f_{P}+c_{0},$$where *c_i_* are calibration coefficients. Applying Equation (11) to the simulation and taking into account a typical gauge factor of FBGs equal to 1.2 pm of wavelength shift per microstrain applied to the fiber [40], the central frequency of the AFBS is calculated for five cases: without strain, at *t* = 6.35 s from the beginning of the maneuver for both front left and front right bearings, and at *t* = 6.8 s for both front left and front right bearings. The diagram showing the relative positions of the AFBS spectra and the filter with an inclined frequency response for the abovementioned cases is presented in Figure 10a.

For each of the five cases of the AFBS central frequency, the spectra of the electrical signal at the photodetector output were simulated. Figure 10b presents the spectra for each corresponding case. As demonstrated in Figure 10b, the amplitude of the electrical signal of the photodetector at the address frequency monotonically increases with the increase of the AFBS central frequency shift.

## 7. Experimental Results

The prototype bearing depicted in Figure 11a was instrumented with a single AFBS-sensor with the address frequency of 6.05 GHz. In order to ensure the uniformity of sensor strain and to eliminate lateral deformation of the sensor (since large lateral deformations applied to FBG impair the accuracy of the standard frequency-shift–strain relation of sensors [41]), a notch corresponding to the sensor length was made on the outer ring of the bearing. For preliminary testing, a static load was applied to the bearing by means of a mechanical press shown in Figure 11b, and the inner ring of the bearing was rotated using an electric screwdriver with the rotational speed of 360 rpm. The resulting amplitude of the measuring channel relative to the reference channel is presented in Figure 11c.

For the dynamic testing, the experimental setup included a B-class passenger vehicle with a prototype load-sensing bearing installed in the modified front right wheel hub with a notch to accommodate the sensor. The sprung mass of the testing vehicle was approximately 1200 kg, which is close to the one of the simulated vehicle.

The conditions of the split-mu braking test were unavailable at the time when experimental studies were carried out; therefore, straight-line braking to a standstill from the initial velocity of 30 km/h was chosen as a testing procedure. The testing was performed on a private driveway with dry asphalt pavement and air temperature of 21 °C.

Figure 12 illustrates the results of the tangential strain estimation for the whole duration of the testing maneuver. As can be seen, the obtained experimental values are comparable with the simulation data. Higher experimental values of strain are mainly caused by higher road traction during the testing in comparison with the simulated conditions.

The experimental results confirm the applicability of the proposed measurement approach for automotive load-sensing bearings. Further studies are planned to increase the number of sensors used in the system for more comprehensive wheel force measurements. Moreover, due to the temperature sensitivity of the AFBS-based strain sensors, the temperature compensation is required for prolonged testing procedures and real-world operation of the system. This can be realized by means of at least one additional AFBS sensor located in proximity to the other sensors and isolated from strain, which will be considered in further studies.

## 8. Conclusions

The work presents a theoretical and experimental investigation of a novel usage of AFBS in load-sensing bearings for automotive applications. The two-frequency optical spectral response of the AFBS results in a beating signal at the output of the photodetector, the amplitude of which can be used to explicitly define the central frequency shift of the AFBS sensor. According to the simulation results, the estimation of the longitudinal wheel force had an error of 5.44% when the strain data were acquired by a single sensor, with the root-mean-square error of 113.64 N. The experimental results of the strain measurement are fully comparable with the simulation data. Further research will be conducted with an increased number of sensors providing measurements of wheel forces in various directions as well as ensuing temperature compensation for prolonged testing procedures.

## Figures and Tables

**Figure 1 sensors-20-06191-f001:**
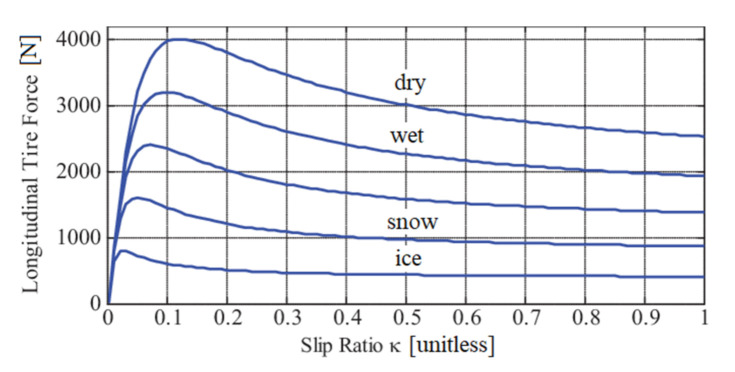
Dependence of longitudinal wheel force on the wheel slip ratio for different conditions of the road surface [25].

**Figure 2 sensors-20-06191-f002:**
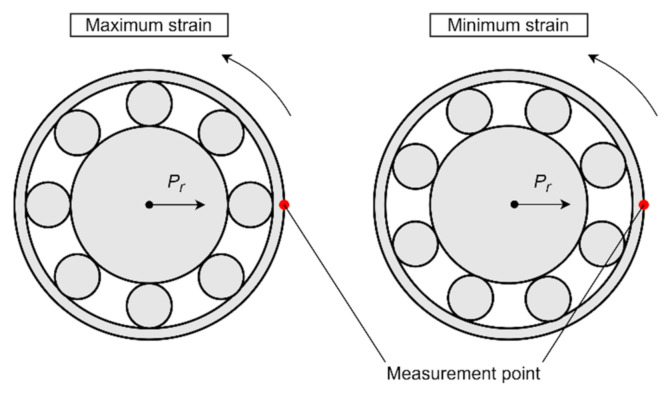
Sensor position relative to the rolling elements at maximum and minimum of measured strain.

**Figure 3 sensors-20-06191-f003:**
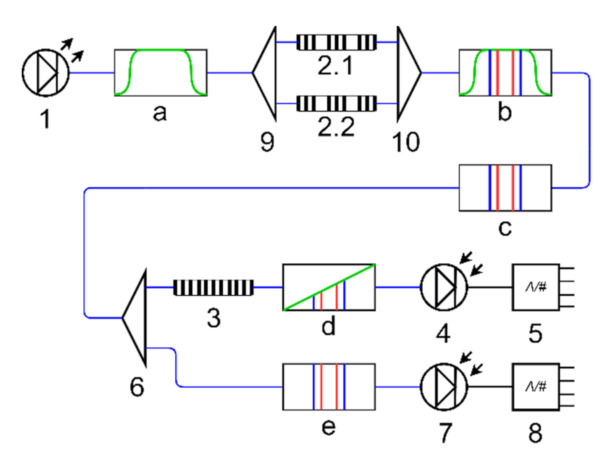
Structure of the interrogation system for two addressed fiber Bragg structure (AFBS)-based sensors: (1) wideband light source, (2.1, 2.2) AFBS sensors, (3) optical filter with a linear inclined frequency response, (4) photodetector of the measuring channel, (5) analog-to-digital converter (ADC) of measuring channel, (6,9) fiber-optic splitters, (7) photodetector of the reference channel, (8) ADC of the reference channel, (10) fiber-optic coupler, (a) spectrum of the wideband light source, (b,c) are the spectra of light propagated through the AFBS sensors, (d) spectra of AFBS sensors at the output of the optical filter, (e) spectra of AFBSs in the reference channel; blue connection lines denote optical fibers, black connection lines represent electrical wires.

**Figure 4 sensors-20-06191-f004:**
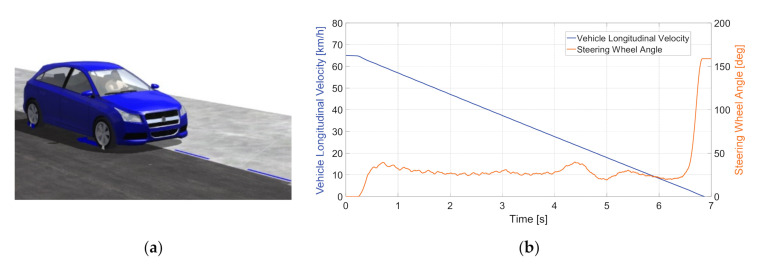
(**a**) Visualization of the split-mu braking test. (**b**) Vehicle longitudinal velocity (blue line) and steering wheel angle (orange line) during the split-mu braking test.

**Figure 5 sensors-20-06191-f005:**
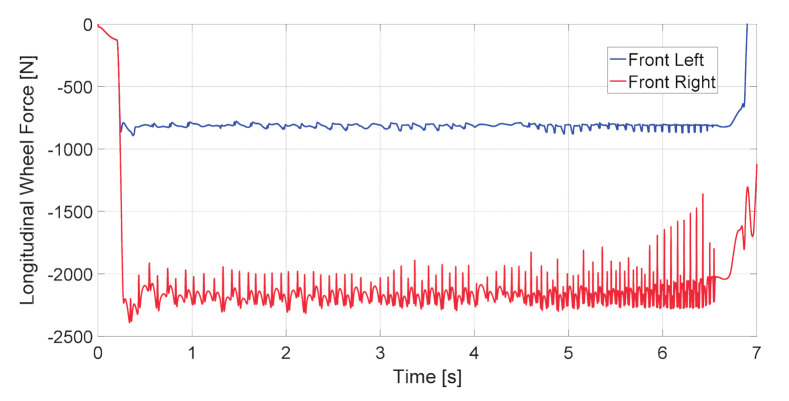
Longitudinal force of the front left wheel (blue line) and front right wheel (red line) during the split-mu braking test.

**Figure 6 sensors-20-06191-f006:**
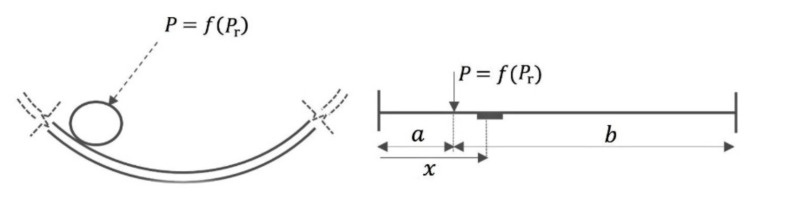
Beam model for a single-load case for estimation of the bearing outer ring deformation [23]: *P* is the load transmitted by a ball, *P*_r_ is the load applied to the inner ring of the bearing, *x* is the sensor position, and *a* and *b* are the load positions relative to the left end and the right end of the beam, respectively.

**Figure 7 sensors-20-06191-f007:**
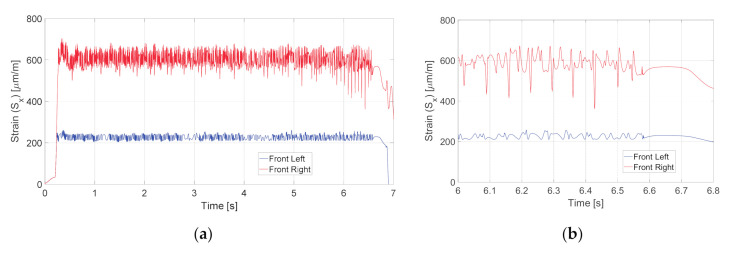
Strain simulated for the front left wheel hub bearing (blue line) and front right wheel hub bearing (red line): (**a**) for the whole duration of the split-mu braking test; (**b**) for *t* = [6, 6.8] s.

**Figure 8 sensors-20-06191-f008:**
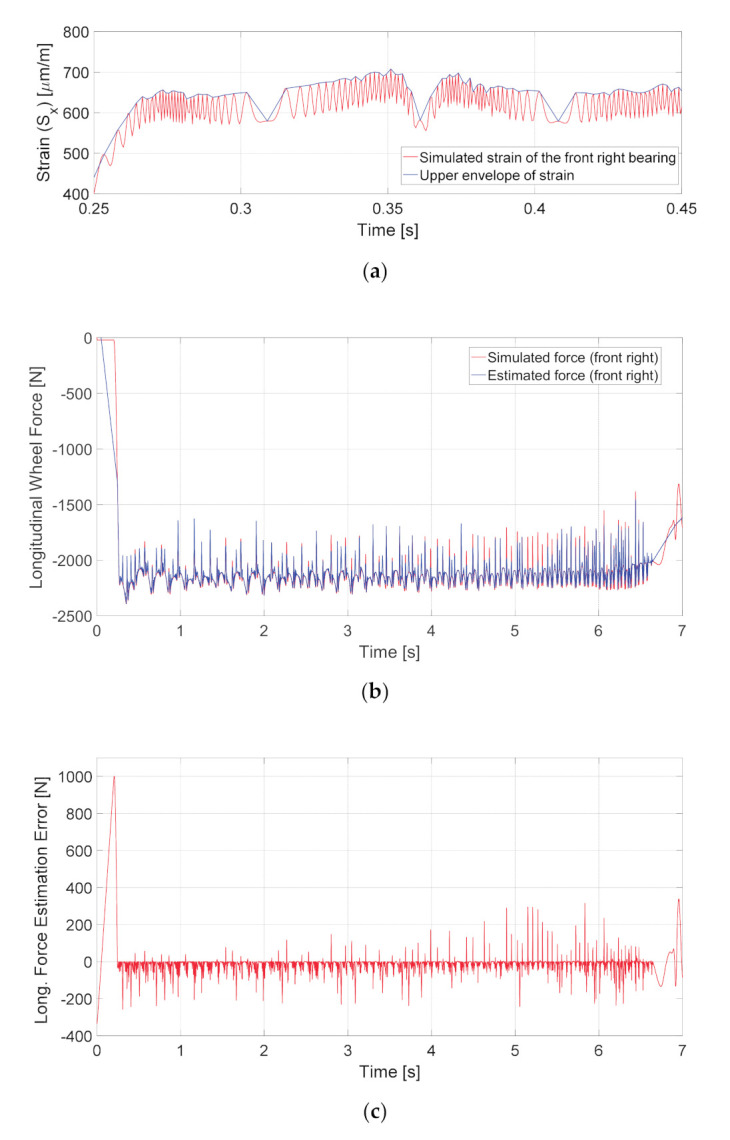
Assessment of the force estimation accuracy by solving the inverse problem: (**a**) strain simulated for the front right hub bearing (red line) and its upper envelope (blue line) shown for a short time interval; (**b**) longitudinal wheel force obtained from the simulation (red line) and estimated force (blue line) for the whole duration of the maneuver; (**c**) force estimation error.

**Figure 9 sensors-20-06191-f009:**
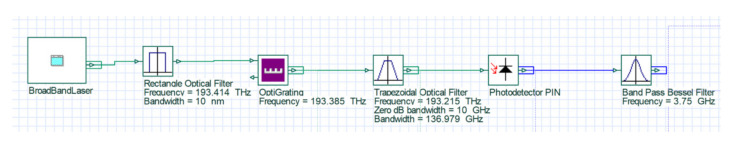
Simulation of the optoelectronic interrogation scheme in the OptiSystem software.

**Figure 10 sensors-20-06191-f010:**
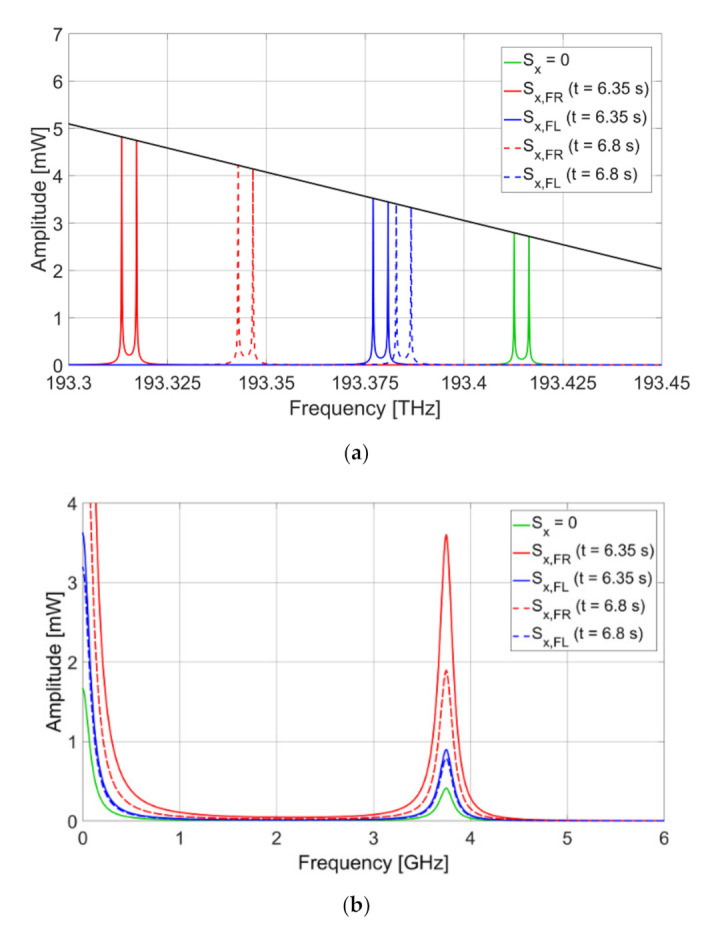
(**a**) AFBS spectra and the optical filter frequency response for five cases of applied strain: AFBS without strain (green line), front right bearing at *t* = 6.35 s (red solid line), front right bearing at *t* = 6.8 s (red dashed line), front left bearing at *t* = 6.35 s (blue solid line), and front left bearing at *t* = 6.8 s (blue dashed line); (**b**) Corresponding spectra of electrical signal at the photodetector.

**Figure 11 sensors-20-06191-f011:**
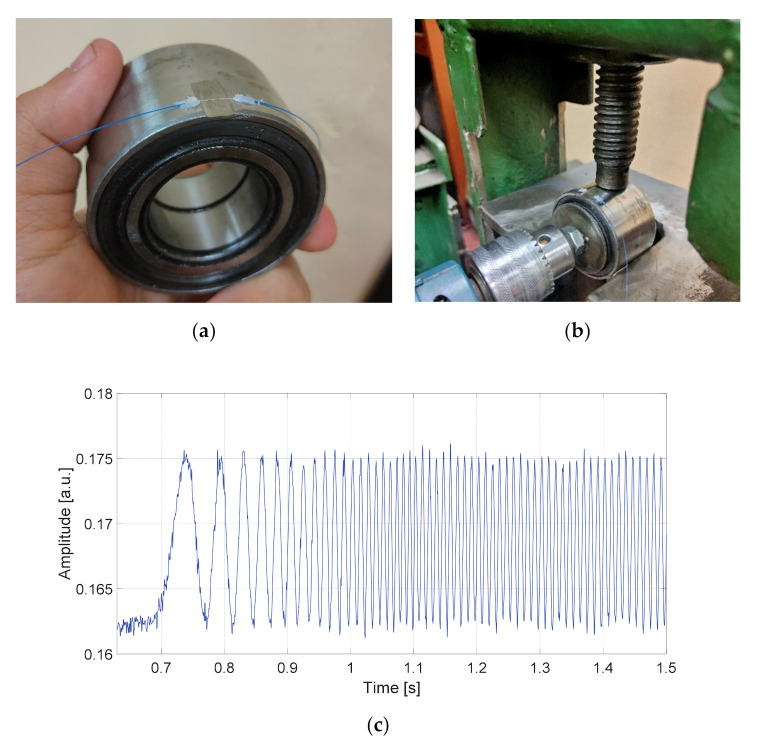
Static load test: (**a**) prototype load-sensing bearing with AFBS sensor; (**b**) experimental setup with a mechanical press for static load test; (**c**) relative amplitude of the resulting beating signal at the photodetector during the static load test.

**Figure 12 sensors-20-06191-f012:**
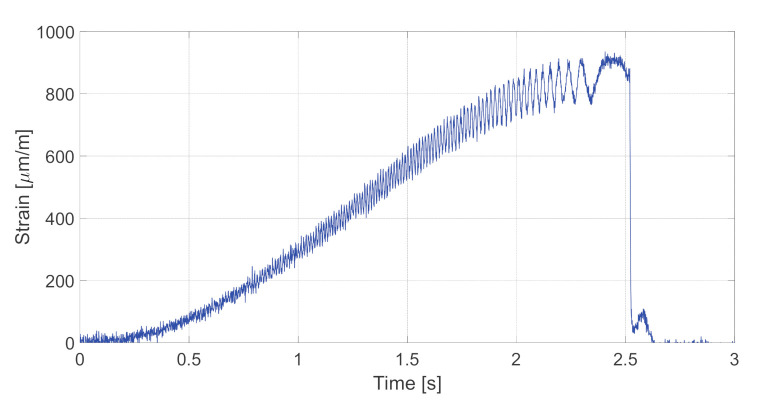
Tangential strain of the bearing measured by AFBS sensor for the whole duration of the dynamic testing procedure.

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
