# Peer review of "Addressed Fiber Bragg Structures in Load-Sensing Wheel Hub Bearings"

_sensors, 2020, doi:10.3390/s20216191_

Round 1

Reviewer 1 Report

This article deals with the measurement of load at wheel hub bearings. The authors use pi-phase shifted fiber Bragg gratings to monitor the load apply on bearings. The applied load is obtained with an interesting interrogation structure that converts directly the variation of the applied load in a electrical varaition given by a photodetector.
Theoritical analysis look in good agreement with the experimental results.
Nevertheless, the authors have assumed tht there was no temperature variation for their simulations or experiments. These are realistic for a resarch work but for measurements in real case the temperature should affect the results. The authors did not give any possible way to mitigate this effect but the thermal sensitivity is ten times bigger than the strain one.
Finally, for me this article cannot be published without any answer concerning the effect of the temperature on the load determination.

Author Response

First of all, we wish to thank the anonymous Reviewer for the precious comments that allowed us to improve the quality of the paper.

The theoretical analysis looks in good agreement with the experimental results.

We thank the anonymous Reviewer for her/his appreciation.

Nevertheless, the authors have assumed that there was no temperature variation for their simulations or experiments. These are really for research work but for measurements in the real cases, the temperature should affect the results. The authors did not give any possible way to mitigate this effect but the thermal sensitivity is ten times bigger than the strain one.

We thank the Reviewer for the precious remark. As we mentioned in the Conclusions (Lines 305–307), further research will include an increased number of sensors to provide temperature compensation for prolonged testing procedures. We have also included the clarification in the Experimental Results section of the revised manuscript (lines 293–297): “Moreover, due to the temperature sensitivity of the AFBS-based strain sensors, the temperature compensation is required for prolonged testing procedures and real-world operation of the system. This can be realized by means of at least one additional AFBS-sensor located in proximity to the other sensors and isolated from strain, which will be considered in further studies.”

Finally, for me, this article cannot be published without any answer concerning the effect of the temperature on the load determination.

We hope that the previously mentioned revisions regarding the temperature compensation are sufficient to make the article accepted for publication.

Reviewer 2 Report

To the authors

The experimental part of the manuscript does not show the dynamic conditions. How was the bearing inserted in the vehicle? What was the position of the FBG relative to the axle? Normally the bearing case involves completely, and all the way round, the bearing. Therefore, there is no space to connect an optical fiber bond to the bearings.

In practice, a thin, fragile, exposed optical fiber, even if carefully installed, will not survive life maintenance of an automobile.

Another concern is that the bearings go tight into the bearing case, which will distribute the efforts caused by the balls when they rotate between the rings. I doubt whether the balls passage will cause any measurable strain due to the effort distribution. In conclusion, in practice the author did not demonstrate that the measured strain will represent the longitudinal force applied to the axle during braking.

For this reason, without a practical application, this work falls into another curious application of an FBG without innovative use. Is my opinion that the paper does not contain innovative substance to be published.

Other comments:

Fig 1: Although being reasonable, is not clear why the longitudinal force on the tires presents a peak for a specific slip ratio. How does the slip ratio interfere with the tire force? Fig 1, that was said to be taken from ref.23, needs more explanation.

L91-101: Terms such as AFBS, 2λ-FBG and 2π-FBG are confused and need a clearer explanation.

Fig. 3: Numbers and letters shown in Fig. 3 must be defined in figure caption.

Fig. 3 confuses electric wires with optical fibers. Symbols used are not accepted internationally.

L103-119: Description of working principle is confused.

Eq.3: Where does Eq. 3 come from?

Fig. 4: Why the steering wheel was changed at the end of the test?

Fig. 6: Definitions of variables shall be presented in the figure caption.

Author Response

We would like to thank the anonymous Reviewer for the comprehensive review of the paper and the valuable remarks.

The experimental part of the manuscript does not show the dynamic conditions. How was the bearing inserted in the vehicle?

We thank the anonymous Reviewer for the remark. As we mentioned in the text (lines 282–283), the experimental part included a straight-line braking to a standstill from the initial velocity of 30 km/h performed with an experimental vehicle equipped with the prototype load-sensing bearing. To our concern, this testing procedure corresponds to the dynamic conditions. Regarding the bearing installation, we stated that “the experimental setup included a B-class passenger vehicle with a prototype load sensing bearing installed in the modified front right wheel hub” (lines 277–278). The modified wheel hub had a notch with the dimensions necessary for the sensor placement. We have added this information in the revised version of the paper (lines 278–279): “…with a notch to accommodate the sensor”.

- What was the position of the FBG relative to the axle?

The position of the FBG relative to the axle is illustrated in Figure 11 (a), where the bearing with the sensor is shown (the vehicle axle coincides with the inner ring axis). The FBG was placed in such a way that it detects the tangential deformation of the outer ring.

- Normally the bearing case involves completely, and all the way round, the bearing. Therefore, there is no space to connect an optical fiber bond to the bearings.

As we stated above, the notch in the modified wheel hub had sufficient dimensions to accommodate the optical fiber with the sensor.

In practice, a thin, fragile, exposed optical fiber, even if carefully installed, will not survive life maintenance of an automobile.

The comment is undoubtedly valid for the bare exposed optical fiber. However, the usage of an armored fiber-optic cable with proper insulation of the sensing elements can provide the sufficient strength and durability of the measurement system comparable to the conventional ABS wheel speed sensor wires.

Another concern is that the bearings go tight into the bearing case, which will distribute the efforts caused by the balls when they rotate between the rings. I doubt whether the balls passage will cause any measurable strain due to the effort distribution.

We thank the Reviewer for the comment. Despite the tight fit of the bearing inside the wheel hub and the effort distribution, the passage of the balls does cause the strain variation of the bearing outer ring, which is confirmed by the experimental results of the current paper, as well as the studies of other researchers (for example, the works referenced as [21, 22] in the article).

- In conclusion, in practice the author did not demonstrate that the measured strain will represent the longitudinal force applied to the axle during braking.

Thank you for the suggestion. The measured strain shown in Figure 12 can be converted into the longitudinal force using the Expressions (6) – (9). However, the graph of the estimated force was not provided since its presence would not be sensible in the current paper due to the lack of reference values of the force, which can only be obtained by the wheel force transducers (similar to the ones presented in the Reference [7]). Unfortunately, the latter was not available during the testing. Nevertheless, the measured values of strain are fully comparable with the simulation data.

For this reason, without a practical application, this work falls into another curious application of an FBG without innovative use. Is my opinion that the paper does not contain innovative substance to be published.

The authors express a hope that the presented clarifications are sufficient to confirm the practical applicability of the approach.

Other comments:

Fig 1: Although being reasonable, is not clear why the longitudinal force on the tires presents a peak for a specific slip ratio. How does the slip ratio interfere with the tire force? Fig 1, that was said to be taken from ref.23, needs more explanation.

We thank the Reviewer for the comment. The behavior of the longitudinal force is explained by the mechanics of tire-road interaction. It is generally accepted that during braking, the contact patch of tire has the adhesive area with good traction and the slipping area, where the traction is reduced. With the increase of slip ratio, the adhesive area becomes smaller, while the slipping area increases. At a specific slip ration, the slipping area becomes prevailing, which leads to the reduction of generated longitudinal force. More detailed discussion of this phenomenon can be found in Reference [41], which was also indicated in the revised article (line 64).

L91-101: Terms such as AFBS, 2λ-FBG and 2π-FBG are confused and need a clearer explanation.

Thank you for the remark. We have changed the structure of the paragraph and added the following explanation (91–96): “An AFBS is a fiber Bragg grating, which has the optical frequency response with two narrowband components. The frequency spacing between the components is called the address frequency and lies the radio frequency range. The system can include several AFBSs with the same Bragg frequency, and the sensors are distinguished using their address frequencies, which are unique for each sensor in the system and do not change when the sensors are subjected to strain or temperature fields.”

Fig. 3: Numbers and letters shown in Fig. 3 must be defined in figure caption.

We thank the Reviewer for the suggestion. The figure caption was modified in the revised article (lines 120–126): “…(1) is a wideband light source, (2.1) and (2.2) are the AFBS sensors, (3) is the optical filter with a linear inclined frequency response, (4) is the photodetector of the measuring channel, (5) is the ADC of measuring channel, (6) and (9) are the fiber-optic splitters, (7) is the photodetector of the reference channel, (8) is the ADC of the reference channel, (10) is the fiber-optic coupler; (a) is the spectrum of the wideband light source, (b) and (c) are the spectra of light propagated through the AFBS sensors, (d) is the spectra of AFBS sensors at the output of the optical filter, (e) is the spectra of AFBSs in the reference channel.”

Fig. 3 confuses electric wires with optical fibers. Symbols used are not accepted internationally.

Thank you for the recommendation. The symbol of the optical source was changed in Figure 3. Unfortunately, the authors did not find the internationally accepted symbol for a fiber-optic coupler. Nevertheless, we have used the common symbol used in various works of other researchers. Moreover, as it was indicated in the previous remark, we have included the description of all the components in the figure caption. The optical fibers in the revised Figure 3 have been denoted as blue lines, while electrical wires are represented in black, which is also indicated in figure caption (lines 126–127): “…blue connection lines denote optical fibers, black connection lines represent electrical wires”.

L103-119: Description of working principle is confused.

Unfortunately, it is not clear from the remark what exactly is confused in the description of the working principle. The description was carefully checked by the authors. As far as we are concerned, this description fully corresponds to the working principle.

Eq.3: Where does Eq. 3 come from?

The Equation 3 is introduced by the authors to represent the optical spectral response of an addressed fiber Bragg structure. This simplified formulation represents each transparency window of an AFBS as a harmonic in a complex exponential form.

Fig. 4: Why the steering wheel was changed at the end of the test?

Thank you for the question. The steering wheel angle was rapidly increased at the end of the test due to the behavior of the virtual driver model used in the CarSim software by default.

Fig. 6: Definitions of variables shall be presented in the figure caption.

The authors thank the Reviewer again for the suggestion. The figure caption has been modified in the revised paper (lines 179 – 182): “…P is the load transmitted by a ball, Pr is the load applied to the inner ring of the bearing, x is the sensor position, a and b are the load positions relative to the left end and the right end of the beam, respectively”.

Reviewer 3 Report

The paper is well organized in terms of literature review, analytical & numerical simulation, test setup, and experimental study. The proposed work has promising applications and is scientifically sound. Here are some of my comments on the paper.

- Regarding the definition of error in the longitudinal wheel force

The solutions from the analytical and numerical analyses are compared in this work. And the discrepancy is described as an error. Does it mean that one of the analyses can serve as the correct solution? Or should those two be considered as two of the possible solutions?

- Optical interrogation system in Figure 3.

It will be easier for casual readers to understand the signal processing if some representative spectra of signals at different stages are added. The terms "wavelength" and "frequency" are used in the papers for optical signal processing. In some cases, the conversion process of those domains is not described. Please add an explanation when those conversions happen during signal processing.

- The discrepancy between the simulation and experimental results of the tangential strain

The envelope of the simulation and experimental results seem quite different. The sustained strain regions in the simulation result between 0.5 to 6.5 seconds are missing from the experimental results. Please explain why the two results' envelopes differ.

Author Response

We would like to thank the anonymous Reviewer for appreciating the paper and the valuable remarks and suggestions.

The paper is well organized in terms of literature review, analytical & numerical simulation, test setup, and experimental study. The proposed work has promising applications and is scientifically sound. Here are some of my comments on the paper.

The authors thank the anonymous reviewer for his/her appreciation.

- Regarding the definition of error in the longitudinal wheel force

The solutions from the analytical and numerical analyses are compared in this work. And the discrepancy is described as an error. Does it mean that one of the analyses can serve as the correct solution? Or should those two be considered as two of the possible solutions?

We thank the Reviewer for the comment. The aim of the approach discussed in lines 208–218 is to define the possible method error of the force estimation based on the measured bearing strain. We use the analytical model of the bearing deformation presented in Equations (6) – (9) to emulate the ‘correct’ values of strain with periodic variation due to the rotation of the balls in the bearing. In practice, it is not always possible to define if the local maximum or minimum of measured strain results from the position of the balls or it is caused by the wheel load fluctuation, which is present in the simulated vehicle. Therefore, when estimating the force from the measured strain, we propose to take into account only the local maxima of strain and to calculate the load assuming that the condition of maximum strain (Equations 6 and 7) is valid.  

- Optical interrogation system in Figure 3.

It will be easier for casual readers to understand the signal processing if some representative spectra of signals at different stages are added. The terms "wavelength" and "frequency" are used in the papers for optical signal processing. In some cases, the conversion process of those domains is not described. Please add an explanation when those conversions happen during signal processing.

Thank you for the valuable remarks. The authors have already included the corresponding optical signals’ spectra in Figure 3 (diagrams a–e). The interrogation principle is further illustrated in Figure 10, where the optical spectra (a) are supplemented with the corresponding spectra of the electrical signal at the photodetector (b). In the revised paper, the term “wavelength” has been replaced with the term “frequency” to ensure consistency throughout the text (lines 17, 88, 98, 106, 116, 142, 234, 236, 237, 238, 240, 244, 251, 260, 263, 268, 302; Figures 9 and 10 (a); Equations 10 and 11).

- The discrepancy between the simulation and experimental results of the tangential strain

The envelope of the simulation and experimental results seem quite different. The sustained strain regions in the simulation result between 0.5 to 6.5 seconds are missing from the experimental results. Please explain why the two results' envelopes differ.

The mentioned discrepancy between the simulation results (Figure 7) and the experimental results (Figure 12) is due to the difference between the drivers’ inputs and the road conditions. The simulated vehicle performed braking on a slippery road. It quickly reached the limit of its wheel traction, which resulted in the sustained strain regions, and the ABS was activated. On the other hand, due to the available conditions, the experiment was carried out on dry asphalt, and the wheels of the vehicle did not reach the traction limit, therefore, the wheel force and the corresponding strain did not have the sustained region.

Round 2

Reviewer 2 Report

No more comments. I have listed all my concerns on the first review.

Author Response

The authors thank the Reviewer for the valuable comments that she/he provided in the previous review. The authors have addressed the remarks in the new version of the manuscript and will take them into account in further research.